# New Approaches for Cryptococcosis Treatment

**DOI:** 10.3390/microorganisms8040613

**Published:** 2020-04-23

**Authors:** Cristina de Castro Spadari, Fernanda Wirth, Luciana Biagini Lopes, Kelly Ishida

**Affiliations:** 1Laboratory of Antifungal Chemotherapy, Department of Microbiology, Institute of Biomedical Sciences, University of São Paulo, São Paulo 05508-000, Brazil; spadaricris@gmail.com (C.d.C.S.); fwirth@gmail.com (F.W.); 2Laboratory of Nanomedicine and Drug Delivery Systems, Department of Pharmacology, Institute of Biomedical Sciences, University of São Paulo, São Paulo 05508-000, Brazil; lublopes@usp.br

**Keywords:** antifungal, blood–brain barrier, central nervous system, *Cryptococcus*, cryptococcal meningitis, drug repurposing, drug delivery systems, nanocarriers, nanotechnology, synthetic molecules

## Abstract

Cryptococcosis is an important opportunistic infection and a leading cause of meningitis in patients with HIV infection. The antifungal pharmacological treatment is limited to amphotericin B, fluconazole and 5- flucytosine. In addition to the limited pharmacological options, the high toxicity, increased resistance rate and difficulty of the currently available antifungal molecules to cross the blood–brain barrier hamper the treatment. Thus, the search for new alternatives for the treatment of cryptococcal meningitis is extremely necessary. In this review, we describe the therapeutic strategies currently available, discuss new molecules with antifungal potential in different phases of clinical trials and in advanced pre-clinical phase, and examine drug nanocarriers to improve delivery to the central nervous system.

## 1. Introduction

*Cryptococcus* infection and development of cryptococcosis in humans generally occurs due to inhalation of yeasts or spores of *Cryptococcus neoformans* and *Cryptococcus gattii* present in the environment [1,2]. *Cryptococcus* causes pulmonary cryptococcosis, and in some patients, the infection may remain latent or oligosymptomatic for a long period. After pulmonary infection, most cases evolve to hematogenous dissemination, with a special predilection for the central nervous system (CNS) leading to cryptococcal meningitis (~90%); in addition, the occurrence of lesions in other tissues is a serious sign of fungus spread [1,3,4].

As a common opportunistic infection in patients with advanced HIV infection, cryptococcosis is the leading cause of meningitis accounting for ~223,100 cases/year, and over 81% mortality in the world [5]. Cryptococcosis-related deaths are most frequent in the sub-Saharan Africa, where mortality reaches 70% [6]. Although access to antiretroviral therapy has resulted in a substantial reduction in the incidence of meningitis by *Cryptococcus* in high-income countries, this infection is likely to remain a major cause of HIV-related mortality in low- and middle-income countries, where antiretroviral treatment is insufficient/unavailable and begins at an advanced stage of HIV/AIDS [7,8].

The antifungal treatment depends of the cryptococcosis clinical form and immunological state of the patient [9,10]. The current antifungal arsenal available for cryptococcosis treatment is limited to three drugs, used alone or in combination: Amphotericin B deoxycholate (AMB) and its lipid formulations (liposomal AMB (LAMB), AMB lipid complex (ABLC), and AMB colloidal dispersion (ABCD)), flucytosine (5-fluorocytosine or 5-FC), and fluconazole (FLC) [9]. In addition to the limited therapeutic options, high attendance and recurrence rates due to the increased resistance of *Cryptococcus* to FLC and 5-FC have been reported [11,12].

Treatment of CNS infections is often difficult because the blood–brain barrier (BBB) limits the diffusion of molecules to the brain tissues, and efflux pumps reduce drug concentrations in the CNS [13]. To gain access to the CNS, drugs can also pass through tight junctions that are much smaller in the BBB (20 Å) than in other tissues of the organism (100 Å), which prevent the diffusion of drugs with high molecular weight (MW). The upper MW limit for efficient crossing of BBB is 400–500 g/mol [14,15], and beyond that, higher lipophilicity and volume of distribution are important properties associated with maximal CNS exposure [16].

Among the antifungals available for the treatment of cryptococcal meningitis, 5-FC (MW = 120 g/mol) and FLC (MW = 309 g/mol) diffuse more easily and present excellent cerebrospinal fluid (CSF) and brain tissue penetration (52–100% of serum concentration) [14,16]. In contrast, AMB is composed of large molecules (MW = 924 g/mol), and although AMB deoxycholate and lipid formulations (ABLC and LAMB) have been previously associated with low penetration in the CSF and brain, the antifungal therapy with these formulations resulted in clinical success [16]. Interestingly, LAMB showed lower penetration in the brain tissue than the AMB deoxycholate formulation (3% vs. 27%) [16].

This scenario emphasizes the pressing need for new strategies and alternatives for the antifungal treatment of cryptococcosis, especially the meningitis. In this review, we describe the conventional therapy of cryptococcosis and the main characteristics of the antifungals currently used; and we discuss new antifungal molecules with anti-*Cryptococcus* activity potential and nanocarrier-based formulations to aid antifungals penetration in the CNS.

## 2. Current Therapy

The treatment of cryptococcal meningitis consists of three phases: induction (2 weeks), consolidation (8 weeks) and maintenance (6–12 months). The guidelines of the Society for Infectious Diseases of America [9] and the World Health Organization [17] emphasize the importance of the use of potent fungicidal drugs during the induction phase; however, worldwide access to antifungal drugs is still inadequate [18], which highlights the importance of alternative treatment strategies.

The primary therapy of cryptococcal meningitis depends on the condition of the patients infected with *Cryptococcus.* For HIV-infected, HIV-non infected and non-transplanted individuals, the primary therapy consists on the induction with AMB (0.7–1.0 mg/kg/day) plus 5-FC (100 mg/kg/day) for 2 weeks. For consolidation and maintenance, FLC at 400 mg/day for 8 weeks (minimum) and at 200 mg/day for 6–12 months, respectively, are employed. In addition, there are other alternative regimens; for example, in case of AMB intolerance, LAMB (3–4 mg/kg/day) or ABLC (5 mg/kg/day) can be used. If 5-FC is not used, AMB deoxycholate or AMB lipid formulations should be maintained for at least 2 weeks [9].

For patients with nonmeningeal cryptococcosis forms as pulmonary (immunosuppressed and nonimmunosupressed) and nonpulmonary cryptococcosis, FLC (400 mg/day) for 6–12 months is recommended. For the pulmonary (nonimmunosupressed) form, voriconazole (VRC) (200 mg twice/day), itraconazole (ITC) (200 mg/day), and posaconazole (POS) (400 mg twice/day) are acceptable alternatives if FLC is unavailable or contraindicated [9].

### 2.1. Amphotericin B (AMB)

Despite AMB dose-limiting toxicity, it has remained the gold standard for treating disseminated life-threatening fungal infections [19]. Its fungicidal effect is associated with AMB binding to ergosterol in the membranes of fungal cell (Figure 1) [20,21]. AMB perturbs the membrane function, causing leakage of cellular contents, and leads to death by cellular dysfunction [20,21]. The first commercially available formulation was Fungizone^®^, a conventional micellar form of AMB and deoxycholate. Currently, parenteral formulations based on lipid carriers are also available, and they include a liposomal formulation (LAMB), a lipid complex formulation (ABLC), and a colloidal dispersion (ABCD) [22]. Their main advantage is the reduction of side effects of AMB [23].

Resistance to AMB is rare and often caused by a decrease in the amount of ergosterol in the plasma membrane or a change in the target sterol, which leads to a decrease in the binding of AMB [19,24]. Some fungal cells have a mutation in the ergosterol biosynthesis pathway, producing ergosterol-like compounds instead of ergosterol, which have lower binding affinity for AMB [25,26].

### 2.2. Flucytosine (5-FC)

5-FC was synthesized in 1957 as a potent antitumoral agent [27,28]. 5-FC is taken into the fungal cell by cytosine permease, and its action as an antifungal agent depends on its conversion to 5-fluorouracil (5-FU) within of the target cells. 5-FU becomes incorporated to the RNA and inhibits DNA synthesis by thymidylate synthase inhibition (Figure 1). It is most active agent against yeasts, including *Candida* and *Cryptococcus* spp. [28]; however, the occurrence of resistance to 5-FC prevents its use as a single agent [28,29,30,31,32]. Currently, its use is indicated only in combination with other antifungals, mainly AMB [23,28]. 5-FC exhibits significant adverse effects, in particular hepatotoxicity and myelotoxicity, which is probably due to toxic 5-FU plasma concentrations.

### 2.3. Fluconazole (FLC)

FLC is a triazole agent that inhibits the fungal cytochrome P450-dependent lanosterol C14-alpha-demethylase (Erg11 or Cyp51) leading to ergosterol biosynthesis inhibition (Figure 1) [12,33,34]. FLC diffuses easily into the cerebrospinal fluid, sputum and saliva, and is concentrated in the urine and skin [35]. The most frequent adverse effects are gastrointestinal events, headache and skin rashes; isolated instances of clinically evident hepatic dysfunction have occurred in patients with AIDS [36]. Over the years, there has been a gradual increase of resistance to FLC in clinical isolates of *C. neoformans,* and nowadays, resistance is a relatively common event in relapse episodes of cryptococcal meningitis [12]. FLC resistance phenotype in *Cryptococcus* spp. have been associated with mutations in the *ERG11* gene [12,37,38]. However, heteroresistance in *Cryptococcus* spp. can lead to higher FLC tolerance by selection of heteroresistant clones after induction due to previous exposure to FLC [12,39].

### 2.4. Voriconazole (VRC)

VRC was developed to increase the antifungal spectrum of available triazoles. To reach this goal, the molecule of FLC was modified, with the substitution of the fluoropyrimidine ring for one of the azole groups, and addition of the α-methyl group to provide fungicidal activity against molds [40,41]. The most frequently reported adverse effect of VRC is transient visual disturbances, that are often associated with higher doses, and considerable hepatotoxic effects. In addition, studies reported important drug interactions with VRC [41]. An ongoing clinical trial study (from 2020 to 2022), named “Three Induction Treatments on Cryptococcal Meningitis (TITOC)”, is investigating its use for cryptococcosis treatment at the Hospital of the University of Zhejiang, China (NCT04072640, www.clinicaltrials.gov). Resistance to this azole is not a common event, but there are reports in the literature in recent years [42,43,44,45].

## 3. New Molecules and Drug Repurposing

Immunobiological and new molecules acting on non-conventional targets or other structures of the fungal cell might have potential as antifungal agents. In this context, drug repositioning is an interesting strategy for antifungal discovery because pharmacokinetics and safety data in humans have been previously established. Therefore, expanding the application of a drug to additional diseases is both cost and time-effective [46,47]. In this section, we will discuss new molecules and drugs currently used in the treatment of other diseases that have activity against *Cryptococcus* spp. (Figure 1).

### 3.1. Interferon-Gamma (IFN-γ)

IFN-γ is an endogenous cytokine with several biological properties and activities, including a key role in the host response to intracellular pathogens, directing the immune system towards the protective Th1 type immunity [48]. Exogenous IFN-γ has been investigated as a potential adjunct agent for treatment of cryptococcal meningitis. In a murine model of pulmonary and disseminated infection, IFN-γ administration resulted in the decrease of the fungal burden in the infected organs, and significantly extended mice survival [49]. One phase II clinical trial (NCT00012467) suggested that IFN-γ may provide rapid and early sterilization of CNS in patients with HIV-associated cryptococcal meningitis without pronounced adverse effects [48]. However, in another study, it was observed that even though administration of IFN-γ improved the fungal clearance from the CNS, it failed to significantly decrease patient mortality [50].

### 3.2. Mycograb

Mycograb is a recombinant human antibody against fungal heat shock protein 90 (Hsp90), which are chaperones required for the maintenance of cellular homeostasis in various fungal pathogens [51,52]. *Cryptococcus neoformans* isolates were susceptible to mycograb at MIC values from 256 to 1024 µg/mL, and a synergistic effect was observed in combination with AMB [53]. The efficacy and safety of mycograb for cryptococcal meningitis are under evaluation in ongoing phase II clinical trials (NCT00324025 and NCT00847678).

### 3.3. 18B7

18B7 is a monoclonal antibody directed against the capsular polysaccharide of *C. neoformans.* Administration of 18B7 promoted rapid clearance of serum cryptococcal antigen and deposition in the liver and spleen, and presented no reactivity with normal mouse, rat, or human tissues [54]. It also reduced the fungal burden in tissues, improved granuloma formation, and demonstrated synergism with AMB, FLC and 5-FC in mice [55,56,57,58]. 18B7 was evaluated in a phase I clinical trial in HIV-infected patients with cryptococcal meningitis, being well tolerated in doses up to 1 mg/kg without evidence of toxicity [59].

### 3.4. APX001 (Fosmanogepix)/APX001A (Manogepix)

APX001 (prodrug of APX001A) is a first-in-class small-molecule antifungal drug candidate that inhibits the fungal enzyme Gwt1 (an inositol acylase) in the glycosylphosphatidylinositol (GPI) biosynthesis pathway [60]. The APX001A MIC ranged from 0.03 to 2 µg/mL for 48 *Cryptococcus* spp. clinical isolates in vitro [61,62], and demonstrated in vitro synergism with FLC [62]. APX001 alone or in combination with FLC decreased the fungal burden in the lungs and brain using cryptococcal meningitis murine model [62]. Other structural analogues of APX001A also demonstrated an excellent in vitro inhibitory effect on *C. neoformans*; and using in vivo assay APX2096 (prodrug of APX2039) led to a nearly complete or complete sterilization of lungs and brain [62]. The preclinical efficacy of APX001/APX001A against *Cryptococcus* associated with previous safety and pharmacological data (NCT02957929 and NCT02956499) lend support to further clinical evaluation of the molecule for treatment of human cryptococcosis.

### 3.5. T-2307

T-2307 is a novel arylamidine derivative with broad-spectrum of action and potent in vitro and in vivo activities, that acts by selectively disrupting mitochondrial function in yeasts [63]. The antifungal activity for *C. neoformans* was observed at MIC ranging from 0.0039 to 0.0625 µg/mL [64], and for *C. gattii* at 0.0078–0.0625 µg/mL [65]. The efficacy of T-2307 was confirmed in murine models of cryptococcosis: at 0.1 mg/kg, T-2307 significantly delayed mortality in mice infected by *C. neoformans* when compared with the untreated group, and T-2307 exhibited a superior protective effect compared to AMB at similar treatment regimens [64]. Administration of T-2307 alone at 2 mg/kg/day significantly reduced viable cell counts in the lungs and brain of mice infected by *C. gattii* and the results were similar to standard treatments [65].

### 3.6. Sertraline

Sertraline is an antidepressant that belongs to the group of selective inhibitors of serotonin reuptake. Initially used for treatment of major depressive disorder, it is now also approved for management of obsessive-compulsive, panic and post-traumatic stress disorders [66]. Although its mechanism of action on fungi has not fully elucidated, inhibition of protein synthesis in *Cryptococcus* spp. has been described [67]. In vitro studies showed that sertraline is effective to inhibit *Cryptococcus* growth at 1–8 µg/mL; in contrast to FLC, sertraline showed fungicidal effect at concentrations higher than 6 µg/mL [67,68]. Murine cryptococcosis model confirmed the antifungal activity observed in vitro, in which sertraline at 15 mg/kg decreased the fungal burden in the brain and spleen when compared with the untreated group [67,68]. Sertraline combined with FLC in vitro showed either additive or synergistic effects, and in animal models, this drug combination led to fungal clearance at a greater rate than either drug alone [67,69,70]. Sertraline use for cryptococcal meningitis treatment alone or in combination with AMB and FLC was investigated in phase III clinical trials (NCT01802385 and NCT03002012), and these studies demonstrated that sertraline did not reduce the mortality rate of patients. This lack of efficacy appears to be multifactorial, and might be associated with insufficient duration of therapeutic sertraline concentrations [71].

### 3.7. Tamoxifen

Tamoxifen belongs to the pharmacological class of selective estrogen receptor modulators; it is an estrogen receptor agonist in the bone, cardiovascular system, and endometrium, while acting as an estrogen receptor antagonist in the breast tissue. This drug is clinically used to treat and prevent breast cancer and osteoporosis [72]. Tamoxifen has in vitro antifungal activity against *Cryptococcus* spp. clinical isolates, with MIC ranging from 2 to 16 µg/mL, acting synergistically when combined with AMB and FLC [73,74]. In the murine disseminated cryptococcosis model, treatment with tamoxifen at 200 mg/kg/day combined with FLC at 5 mg/kg/day decreased the burden fungal by ~1 log in the brain tissue [74]. The authors of the study suggested the use of this drug for treatment of cryptococcosis because high concentrations (well above of the MIC values) were reached in the CNS in addition to the antifungal activity inside macrophages, synergism with existing therapies AMB and FLC, and good oral bioavailability [72,74]. At the moment, clinical trials (phase II) are being carried out to evaluate the efficacy, feasibility, and safety of tamoxifen in combination with standard therapies (AMB and FLC) in the treatment of cryptococcal meningitis (NCT03112031). Although tamoxifen activity against *Cryptococcus* has been reported, and the drug is under evaluation in ongoing clinical trials for cryptococcosis treatment, its mechanism of action has not been elucidated yet.

### 3.8. AR-12

AR-12, a small molecule derived from celecoxib, was tested as an antitumoral agent in phase I clinical trials, and licensed to Arno Therapeutics (NCT00978523) [75]. AR-12 is a non-nucleoside acetyl CoA synthetase inhibitor as previously investigated in *S. cerevisiae* and *C. albicans* [76]. This molecule has broad-spectrum antifungal activity, including for *C. neoformans*, with MIC value of 4 µg/mL, and AR-12 was demonstrated to be effective in a murine model of disseminated cryptococcosis when combined with FLC (dose at 100 and 10 mg/kg, respectively), decreasing the fungal burden in the brain [77].

### 3.9. Miltefosine (MFS)

MFS belongs to the alkylphosphocholine class of molecules, and is used in the treatment of cutaneous metastases of breast cancer and leishmaniasis [78]. Studies showed that MFS has a broad-spectrum in vitro antifungal activity, including against *C. gattii* and *C. neoformans* isolates in the both planktonic (0.25–4 µg/mL) and biofilm (8 - ≥16 µg/mL) lifestyles [79,80,81]. Moreover, MFS was effective to control the fungal infection in the larval model of *Galleria mellonella* by *C. gattii* at 10, 20, and 40 mg/kg [82]. MFS at 3.6 and 7.2 mg/kg/day has shown effectiveness in the murine model of disseminated cryptococcosis [80] although this result has been conflicting with other work [83]. This contradiction demonstrates variable translation of in vitro MFS activity to in vivo murine models of disseminated cryptococcosis. Studies evidenced that MFS acts through multiple mechanisms, being able to alter membrane permeability, inhibit phospholipase B1 and induce an apoptotic-like cell death reducing the mitochondrial membrane potential, increasing reactive oxygen species (ROS) production, and inducing DNA fragmentation and condensation [80,81]. Despite the extensive and exciting in vitro reports highlighting MFS usefulness as an antifungal drug, no clinical trial for treatment of fungal infections is under way.

### 3.10. Tetrazoles

Tetrazoles are synthetic molecules produced from azoles, non-metabolized bioisosteric analogs of carboxylic acid and cis-amide; they possess diverse chemotherapeutic properties and are highly selective fungal Cyp51 inhibitor [84,85]. Among tetrazoles, VT-1129 and VT-1598 are more selective for fungal Cyp51 than mammalian Cyp450 enzymes and both molecules showed antifungal efficacy against *Cryptococcus* spp. [86,87]. VT-1129 inhibited the growth of *C. neoformans* and *C. gattii* isolates at 0.003–4 µg/mL and 0.06–8 µg/mL, respectively [88,89]. VT-1598 has lower MIC values (0.06 to 0.15 µg/mL) [90,91]. In addition to the in vitro models, assays using cryptococcosis murine models demonstrated that oral administration of VT-1598 resulted in suitable plasma and brain concentrations, leading to a significant reduction in the brain fungal burden [91]. Recently, phase I clinical trials have started to assess the safety and pharmacokinetics of VT-1598 (NCT04208321).

## 4. Other Molecules and Targets

To find alternatives for cryptococcosis treatment, other molecules have been evaluated in the pre-clinical studies. In vitro screening with some off-patent drugs found 43 drugs capable of inhibiting the growth of *C. neoformans,* such as cliclopirox and auranofin [92]. Other studies have investigated the use of antiprotozoal drugs as benzimidazoles with MIC ranging from 0.125 to 0.45 µg/mL, and flubendazole was found to reduce fungal burden in infected mice [93,94].

Drugs with action on *Cryptococcus* virulence mechanisms have the potential to aid treatment. For example, the herbicide glyphosate inhibited melanization of yeasts at subinhibitory concentrations in vitro, and its administration in mice infected with *C. neoformans* delayed the melanization of yeasts and prolonged mice survival twice as long compared to untreated mice [95]. The vesicular transport system is essential for virulence, being related to the assembly of the mucopolysaccharide capsule, secretion of melanin, lipids, hydrolytic enzymes and other molecules [96]. Two synthetic compounds, N-(3-bromo-4-hydroxybenzylidene)-2-methylbenzohydrazide (BHBM) and 3-bromo-N-(3-bromo-4-hydroxybenzylidene) benzohydrazide (DO), caused accumulation of intracellular vesicles and inhibited surface glucosylceramides, demonstrating antifungal potential for *Cryptococcus* [97].

## 5. Drug Delivery Systems

Drug delivery systems represent a promising alternative for treatment of diseases that affect the brain due to the possibility of enabling drug transport across BBB. They can improve transport by masking physicochemical characteristics of drugs, which circumvents the need for molecular modifications [98]. In addition, these carriers are small in size and can be composed of various materials, such as polymers, lipids, metals and inorganic elements, among others. The most frequently used nanocarriers include liposomes, micelles, polymeric and inorganic nanoparticles, such as gold and silver (Figure 2A) [99,100]. Liposomes are biodegradable colloidal aggregates composed of one or multiple bilayers that encompass an internal aqueous compartment [100]. Nanoparticles are colloidal systems with a compact structure that vary in size from 10 to 1000 nm, in which the drug can be dissolved, dispersed, trapped, encapsulated or attached to a matrix; they can be produced with inorganic or organic materials as PLGA, PLA, PBCA, albumin, among others [101,102].

Nanocarriers may undergo surface modifications to improve drug penetration into the brain, which might involve various processes as represented in Figure 2B,C. Nanocarrier surface modification enables recognition/targeting of specific ligands on the cell surface, leading to cell uptake and transport into the brain after systemic administration via receptor-mediated pathways [103,104]. For example, cationic nanocarriers can interact electrostatically with BBB (which has a net negative charge), and be internalized by adsorption-mediated endocytosis [101]. Other mechanisms that might participate in the ability of nanocarriers to improve transport across BBB include (i) a general surfactant effect, characterized by solubilization and fluidization of the cell membrane, (ii) ability to open tight junctions, and (iii) inhibition of efflux transporters (such as P-glycoprotein) at the BBB, which is largely attributed to the presence of specific surfactants at the nanocarrier surface, such as polysorbate 80 and vitamin E TPGS [105,106]. In addition to polysorbate 80, other compounds have been previously employed to modify the surface of nanocarriers intended for encapsulation of antifungals and treatment of cryptococcal meningitis, including angiopep-2, apolipoprotein E and borneol. In this section, we will discuss nanocarriers with surface modifications aimed at improving transport of antifungal agents across BBB for treatment of cryptococcal meningitis.

AMB is considered the gold standard for cryptococcal meningitis treatment (induction phase) and for other systemic fungal infections despite its multiple adverse effects. Lipid-based nanocarriers (LAMB, ABDC, ABLC) have been shown to reduce AMB toxicity, but not to increase its penetration in the CNS when compared with the deoxycholate (conventional) AMB formulation [16]. This has motivated research for other nanocarriers. A new oral cochleate-amphotericin B formulation (CAMB or MAT2203) has been evaluated in a phase II clinical trial for efficacy and safety in the treatment of cryptococcosis (NCT03196921). Cochleates are generally described as stable phospholipid-cation crystalline structures consisting of a spiral lipid bilayer sheet with no internal aqueous space [107]. In a murine model of infection with *C. neoformans,* oral administration of CAMB delivered AMB at therapeutic levels to the CNS [108]. Moreover, its co-administration with 5-FC showed an equivalent efficacy to deoxycholate AMB with 5-FC, and superior to oral FLC, but with reduced toxicity [108].

Polymeric nanoparticles coated with polysorbate 80, a non-ionic surfactant, have been shown to increase transport through BBB and consequently, improve drug concentrations in the CNS. AMB-poly(lactic acid)-b-poly(ethylene glycol) (AMB-PLA-b-PEG) nanoparticles prepared by nanoprecipitation and coated with polysorbate 80 were able to increase AMB concentration in the brain, suggesting that polysorbate 80 aids transport across the BBB. The formulation significantly reduced the fungal burden in the brain of a mice model of cryptococcosis when compared with deoxycholate AMB, and decreased AMB toxicity [109]. Another type of nanoparticle, poly butyl cyanoacrylate nanoparticles (AMB-PBCA-NP) coated with polysorbate 80 were detected in the brain 30 min after systemic administration and had a higher brain tissue concentration than LAMB. In addition, the survival rate of mice treated with AMB-PBCA- NP (80%) was significantly higher than LAMB (60%) and deoxycholate AMB (0%) [110].

Shao et al. (2010) demonstrated that an angiopep-2 modified PE-PEG (1,2-Distearoyl-sn-glycero-3-phosphoethanolamine-N-[methoxy (polyethylene glycol)-2000]) based micellar drug delivery system loaded with AMB (Angiopep-PEG-PE/AMB) was more effectively transported across the BBB than the deoxycholate AMB and the micelle without angiopep-2 in in vitro and in vivo assays. This superiority was attributed to angiopep-2 being a ligand of low-density lipoprotein receptor-related protein (LRP) present in the brain, which has been reported to mediate the transport of ligands across endothelial cells of the BBB [111]. When Angiopep-PEG-PE/AMB and AMB commercial formulations were compared for treatment of murine cryptococcosis, Angiopep-PEG-PE/AMB was more effective, leading to the highest AMB concentration in the brain, reducing the fungal burden and prolonging the median survival time of mice [112].

ITC and its derivative compound POS, both triazole agents, are acceptable alternatives for treatment of the pulmonary form of cryptococcosis, but not for cryptococcal meningitis due to its difficulty of crossing the BBB [9]. ĆURIĆ and contributors developed poly(butyl cyanoacrylate) nanoparticles for ITC [113], and demonstrated that nanoparticles functionalization with covalent binding of apolipoprotein E enabled targeting of low-density lipoprotein receptor (LDLR) expressed on the endothelial brain capillary cell membrane [114]. ITC was also incorporated in bovine serum albumin nanoparticles (ITC-BSA-NP) modified with borneol (BO) and polyethylene glycol (PEG) (PEG/BO-ITC-BSA-NP) [115]. The nanoparticles significantly increased ITC uptake by the bEnd.3 cells (a mouse brain cell line) and promoted ~2-fold higher brain uptake in mice than ITC conventional formulation [115]. Borneol is a highly lipid-soluble bicyclic monoterpene that can facilitate penetration of drugs in the brain due to BBB opening [116]. Another study used a 29-amino-acid peptide derived from rabies virus glycoprotein (RVG29) conjugated with albumin nanoparticles as carrier for ITC (RVG29-ITC-NP), and demonstrated a significant drug accumulation in the brain compared to albumin nanoparticles without RVG29 and a cyclodextrin formulation of ITC [117].

## 6. Concluding Remarks

Searching for new alternatives for treatment of fungal infections has been pivotal. Here, we highlighted the remarkable deficit of antifungal options in the treatment of cryptococcosis recommended by the current guidelines. Studies have been carried out with the intention of finding new therapeutic alternatives as described in this work; some of these molecules and the use of nanocarriers to improve drug delivery into the CNS may become future therapies for cryptococcal meningitis.

## Figures and Tables

**Figure 1 microorganisms-08-00613-f001:**
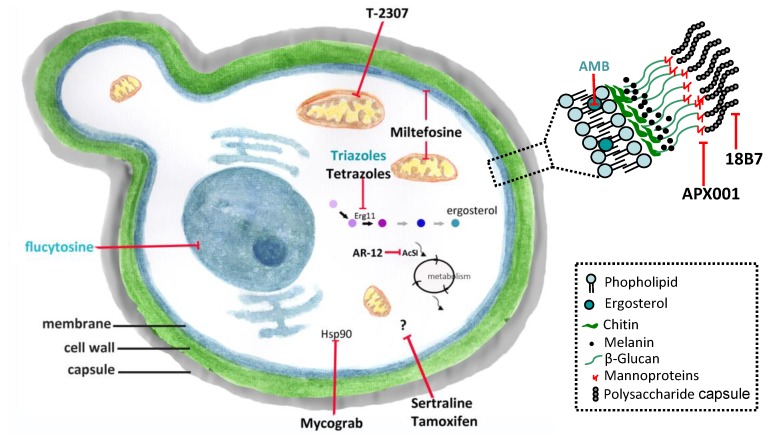
Conventional antifungals and new molecules for cryptococcosis treatment. Amphotericin B (AMB) and azoles inhibit ergosterol and its biosynthesis, respectively, and flucytosine inhibits the nucleic acids synthesis. New molecules acting on non-conventional targets or different structures of fungal cells may have antifungal effects. Erg11 (or Cyp51)—cytochrome P450-dependent lanosterol C14-alpha-demethylase; AcS—Acetyl CoA synthetase; Hsp90—Heat shock protein 90.

**Figure 2 microorganisms-08-00613-f002:**
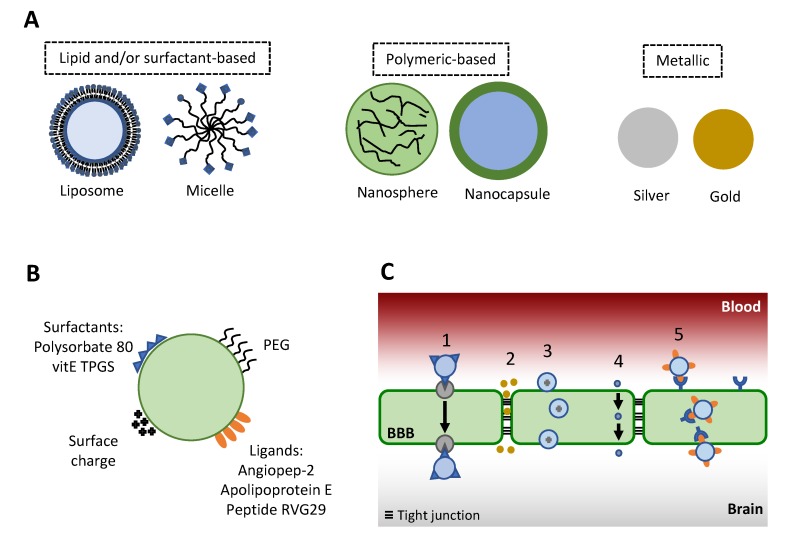
Schematic representation of nanocarriers, their surface modification and possible routes of transport across the blood–brain barrier (BBB). (**A**) Nanocarriers frequently employed for drug delivery to the central nervous system. (**B**) Examples of nanocarrier surface modification to improve passage through the BBB; (**C**) Possible routes of nanocarrier-mediated transport across the BBB. **1**—Nanocarrier-mediated transport; **2**—Paracellular pathway, which can result from the ability of a nanocarrier and/or its components to open tight junctions; **3**—Adsorption-mediated transcytosis; **4**—Transcellular pathway, which might result from the ability of nanocarrier components to improve membrane permeability; **5**—Receptor-mediated transcytosis. PEG: polyethyleneglycol; vitE TPGS: D-α-tocopherol polyethylene glycol 1000 succinate.

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
