# Peer review of "New Approaches for Cryptococcosis Treatment"

_microorganisms, 2020, doi:10.3390/microorganisms8040613_

Round 1
Reviewer 1 Report
This paper reviews the currently available drug treatments for cryptococcosis and discusses new molecules with antifungal potential. The authors also discuss drug nanocarriers as a way to improve drug delivery to the central nervous system. Overall, I think paper reads pretty well and covers many of the molecules with antifungal potential. However, I have some suggestions.
- In the description of each new molecule, could you add anything that is known about the mechanism of action and how they inhibit Cryptococcus? I realize that the mechanism may not be known for some of them, but I think if any of that information is available, it would add to the paper.
- While overall the paper is fairly well written, there are still a number of grammar mistakes. I suggest having the paper edited professionally or having a native English speaker edit the paper.
Minor points:
- On lines 146 and 314, replace “session” with “section”.
- On line 356 “cell” should not be italicized.
Author Response
Response to Reviewer 1 Comments
Dear,
We are thankful for the suggestions, which have undoubtedly contributed to improving our manuscript. We hope that we have addressed all the reviewers’ concerns and that our revised manuscript can be accepted in its present version. All alterations performed in the manuscript are highlighted in yellow.
Please find the point-by-point responses to the concerns raised by the reviewers bellow.
Sincerely,
Point 1: This paper reviews the currently available drug treatments for cryptococcosis and discusses new molecules with antifungal potential. The authors also discuss drug nanocarriers as a way to improve drug delivery to the central nervous system. Overall, I think paper reads pretty well and covers many of the molecules with antifungal potential. However, I have some suggestions.
Response 1: Thank you for your revision and suggestions. All suggestions contributed to improve the paper quality.
Point 2:In the description of each new molecule, could you add anything that is known about the mechanism of action and how they inhibit Cryptococcus? I realize that the mechanism may not be known for some of them, but I think if any of that information is available, it would add to the paper.
Response 2:We have now included the mechanisms of action (or other relevant information about the probable mechanisms) of new molecules as suggested (lines 156 to 158; 211 to 213; 238 to 241; and 261 to 265).
Point 3:While overall the paper is fairly well written, there are still a number of grammar mistakes. I suggest having the paper edited professionally or having a native English speaker edit the paper
Response 3: The manuscript was revised and edited by an English speaker for improvement of work quality. All alterations are highlighted in yellow.
Point 4.Minor points:
- On lines 146 and 314, replace “session” with “section”.
The spelling was corrected.
- On line 356 “cell” should not be italicized.
This was corrected.

Reviewer 2 Report
This was an excellent review on the current therapies for fungal infections with particular emphasis on Cryptococcus spp. Although current therapies rely on three main treatments the authors have provided a nice review on therapeutic alternatives ranging from small molecules to immunotherapy.
In addition, the authors also touched on alternatives to drug delivery with relevance to fungal infections in the central nervous system.
The review is timely albeit a bit a bit short on mode of action but nevertheless it fill an important niche in the medical field.
Author Response
Response to Reviewer 2 Comments
Dear,
We are thankful for the suggestions, which have undoubtedly contributed to improving our manuscript. We hope that we have addressed all the reviewers’ concerns and that our revised manuscript can be accepted in its present version. All alterations performed in the manuscript are highlighted in yellow.
Please find the point-by-point responses to the concerns raised by the reviewers bellow.
Sincerely,
Kelly Ishida
Point 1: This was an excellent review on the current therapies for fungal infections with particular emphasis on Cryptococcus spp. Although current therapies rely on three main treatments the authors have provided a nice review on therapeutic alternatives ranging from small molecules to immunotherapy.
In addition, the authors also touched on alternatives to drug delivery with relevance to fungal infections in the central nervous system.
The review is timely albeit a bit a bit short on mode of action but nevertheless it fill an important niche in the medical field.
Response 1: Thank you for your revision and comments about the paper. According to the suggestion of the Reviewer 1, we have included the mechanisms of action (or other relevant information about the probable mechanisms) of new molecules (lines 156 to 158; 211 to 213; 238 to 241; and 261 to 265). In addition, the manuscript was revised and edited by an English speaker for improvement of work quality.
